# Emerging Evidence of the Significance of Thioredoxin-1 in Hematopoietic Stem Cell Aging

**DOI:** 10.3390/antiox11071291

**Published:** 2022-06-29

**Authors:** Shaima Jabbar, Parker Mathews, Yubin Kang

**Affiliations:** Division of Hematologic Malignancies and Cellular Therapy, Department of Medicine, School of Medicine, Duke University Medical Center, Durham, NC 27710, USA; shaima.jabbar@duke.edu (S.J.); parker.mathews@duke.edu (P.M.)

**Keywords:** hematopoietic stem cells, ageing, thioredoxin, redox systems

## Abstract

The United States is undergoing a demographic shift towards an older population with profound economic, social, and healthcare implications. The number of Americans aged 65 and older will reach 80 million by 2040. The shift will be even more dramatic in the extremes of age, with a projected 400% increase in the population over 85 years old in the next two decades. Understanding the molecular and cellular mechanisms of ageing is crucial to reduce ageing-associated disease and to improve the quality of life for the elderly. In this review, we summarized the changes associated with the ageing of hematopoietic stem cells (HSCs) and what is known about some of the key underlying cellular and molecular pathways. We focus here on the effects of reactive oxygen species and the thioredoxin redox homeostasis system on ageing biology in HSCs and the HSC microenvironment. We present additional data from our lab demonstrating the key role of thioredoxin-1 in regulating HSC ageing.

## 1. Introduction 

Ageing is a natural process in all living organisms and represents a progressive decline in functional activities. Ageing is associated with many pathophysiological disorders including autoimmune diseases, cancers, diabetes, cardiovascular diseases, and neurodegenerative conditions [1]. Hematopoietic stem cells (HSCs) are a rare population of cells that reside in specialized bone marrow (BM) niche and are characterized by their ability to self-renew and differentiate into hematopoietic multipotent progenitors (MPPs) and hematopoietic progenitor cells (HPCs) to maintain hematopoiesis and an immune system during the entire lifespan of the organism [2]. HSCs will age as do all other cells. During ageing, the functionality and integrity of HSCs decline, and various factors such as increased cellular metabolic demands, an altered BM microenvironment, DNA damage, exposure to high levels of free radicals, and epigenetic changes can all contribute to shift HSCs towards ageing phenotypes. The ageing of HSCs is the key process underlying the decline in immune function (so called “immunosenescence”), the lineage skewing of hematopoiesis, and the increase in the incidence of hematological malignancies seen in the elderly population [3] (Figure 1). Currently, the molecular and cellular pathways driving the ageing of HSCs are not well characterized. The continuous existence of this gap in our knowledge poses a significant obstacle to our efforts to attenuate ageing-associated disease and to improve the quality of life for the elderly. In this review, we summarized the biological concepts of HSC ageing including the hallmarks of HSC ageing and the molecular mechanisms that drive HSC ageing. We reviewed the effects of reactive oxygen species (ROS) on HSC ageing, with a particular emphasis on the roles of thioredoxin-1 in HSC ageing. Understanding the mechanisms of HSC ageing and the effects of thioredoxin-1 provides a foundation for us to develop novel approaches to rejuvenate aged HSCs and to attenuate tumorigenesis and thus has important implications in the current era, where there is a critical demographic shift towards an ageing population in the USA. 

## 2. Background: Biological Concepts of HSC Ageing 

HSCs can both self-renew and differentiate to continuously replenish all lineages of blood cells throughout life. In the 1970s and 1980s, scientists postulated that HSCs are ageing-exempt due to their telomerase activity, which might preserve HSC longevity at least beyond the expected human lifespan [4,5]. Our understanding changed when Sean et al. made their landmark discovery, showing that HSCs isolated from older mice exhibited a reduced ability to engraft lethally irradiated mice compared to young HSCs. Although the mechanism behind this biological phenomena was not addressed [6], this seminal observation laid the foundation for our current research addressing the differences between juvenile and aged HSCs.

The ageing of HSCs is complex, and there is no standardized model defining the key mechanisms of HSC ageing. Furthermore, there is no single cell surface marker that can be used reliably to sort and characterize aged HSCs, nor a golden index for the measurement of HSC kinetics and cell cycle dynamics to define the biological age of HSCs. Most of the studies in the field of HSC ageing research have been conducted by enriching HSCs from aged mice and comparing their biological functions to those isolated from young mice.

The hallmark of HSC ageing is their reduced ability for self-renewal and reconstitution potency following serial transplantations in myeloablative recipient mice. Although the number of mouse HSCs increases with age, the ability of aged HSCs to self-renew and reconstitute erythroid and lymphoid lineages in serial transplant recipient mice is impaired. Aged HSCs lose their original compartment signature, and their multilineage differentiation features are reduced due to stress and exhaustion [7,8,9,10]. Aged HSCs show lineage skewing with more myeloid differentiation and less lymphoid cell production. Using an inducible Fgd5-based HSC lineage tracing model, scientists have found that HSCs usually preferably differentiate to the myeloerythriod lineage and less to the lymphoid (adaptive immunity) lineage. In the setting of acute thrombocytopenia, HSCs rapidly differentiate into the platelet lineage [8,10,11,12,13]. The differentiation kinetic of aged HSCs seems to be defective. In fact, in old mice and humans, HSCs increase their proliferation and expansion frequency and skew their differentiation to the myeloid lineage (and in some cases, myeloid malignancies) and less to the erythroid and lymphoid cells [14,15,16]. The lineage skewing of aged HSCs is consistent with the clinical observation that hematology disorders such as anemia and hematological malignancies such as leukemia, myeloma, and lymphoma are linearly correlated with lifespan [17,18,19,20,21]. 

Aged HSCs also exhibit defects in stem cell homing and engraftment. It was reported that the homing efficiency of aged HSCs was significantly reduced compared to young HSCs. Liang and his colleague transplanted bone marrow (BM) cells from young or old C57BL/6 mice into old or young Ly5 congenic mice. Interestingly, they found the homing efficiency of HSCs from the old mice (2 years old of age) to be reduced by more than threefold compared to that of HSCs from young mice (2 months of age) [22]. Most of the studies examining the homing efficiency of HSCs involve transplanting BM HSCs into lethally irradiated hosts. These assays test not only the role of HSCs themselves but also the impact of microenvironmental factors on the seeding frequency of HSCs. To dissect out the effect of HSCs themselves versus the impact of the microenvironment, Dykstra and his colleague developed a barcode tool and conducted a co-homing study by labeling old and young BM HSCs with GFP at a fixed ratio and transplanting the HSCs into lethally irradiated or non-irradiated old mice. Their results revealed that the homing frequency of BM HSCs improved in non-irradiated old recipient mice and that old/young HSCs that successfully homed into old recipient mice lost their compartment integrity compared to the pre-homed HSCs [23]. Figure 2 summarizes some of the hallmarks of HSC ageing.

The cellular and molecular pathways driving HSC ageing are complex and remain an area of active investigation. HSCs reside in the BM matrix and are surrounded by non-hematopoietic cells. Factors that are related to HSC ageing are typically classified as intrinsic and extrinsic (microenvironmental) [24]. 

### 2.1. Intrinsic Factors Driving HSCs Ageing 

There are several intrinsic mechanisms that have previously been identified as contributing to ageing-related changes of HSCs. Although much of this discussion will address these mechanisms separately, the intrinsic factors that contribute to HSC ageing are highly interconnected and interdependent.

#### 2.1.1. DNA Damage Response and HSC Ageing 

DNA damage response (DDR) is a cellular intrinsic factor that shifts HSCs towards ageing phenotypes and can be triggered by physical (low dose radiation), chemical (genotoxic agents), or biological insults (replication stress or oxidative stress). During cell cycle arrest, the DNA proofreading enzymes are activated, and DNA double stranded breaks are repaired either by homologous recombination (HR) or by non-homologous end joining (NHEJ) [25]. NHEJ is thought to be error prone and can lead to DNA mutations, while HR is considered as a DNA repair error-free pathway. With each cell division and expansion, there are increased risks for replication stress and cellular proofreading machinery errors, both of which cause DNA damage. Typically, most HSCs reside in a relatively hypoxic niche and are characterized by low metabolic activity and a lower proliferation rate, which make HSCs less likely subject to NHEJ repair [25,26]. However, HSCs residing in the G0 cell cycle are more vulnerable to the NHEJ repair process, resulting in the loss of DNA integrity [26]. Additional lines of evidence suggested that the upregulation of p21 in HSCs in response to DNA damage promotes this faulty DNA repairing system, which may ultimately lead HSCs towards exhaustion and aged phenotypes [27]. 

DNA damage repair mechanisms are related to HSC stemness and differentiation. Through the tumor suppressor p53 pathway, DDR activity can serve as a signal for HSCs to switch from self-renewal to a commitment to differentiate into progenitor cells [28]. p53, as the “cellular gatekeeper”, is essential in coordinating the cellular responses to a broad range of cellular stress factors [29]. Studies suggested that p53 plays an important role in the regulation of HSC quiescence and senescence [27,28], with declining p53 function associated with longevity in naturally aged mice [27,30]. DDR is a key component regulating the long-term self-renewal integrity and differentiation efficiency of HSCs, and the accumulation of DDR through the lifespan can shift HSCs towards ageing phenotypes.

#### 2.1.2. Senescence and HSC Ageing 

In HSCs, cell senescence is characterized by terminal cell cycle arrest in which the cells are unable to undergo self-renewal or differentiation. The clearance of senescent cells using progeroid transgenic mice successfully delayed the progression of ageing-associated disorders [31,32], suggesting that cell senescence plays a major and likely causative role in HSC ageing-related phenotypes. The cell cycle-dependent kinase inhibitors p16 ^INK4a^ and p14(ARF) are important molecules regulating mammalian cell senescence [33,34]. In old murine HSCs, the upregulation of Notch signaling through the activation of the TGF-β/pSmad3 pathway, which led to the inhibition of p16 ^INK4a^, significantly improved the regenerative capacity of aged HSCs [31,32,33,34,35,36]. The administration of fibroblast growth factor 7 in murine models resulted in the repression of p16 ^INK4a^ and the partial rejuvenation of early T cell progenitors [34]. The overexpression of p16 ^INK4a^ in peripheral blood immune cells is positively correlated with human chronological age, suggesting that p16 ^INK4a^ is a potential biomarker of human molecular age [34]. However, there are also studies that suggested a more limited role of p16 ^INK4a^ in the ageing-related changes of HSCs and that the decline of HSC functions during ageing is not dependent on the induction of p16 ^INK4a^ but is instead mediated by other, currently undefined mechanisms [37,38,39]. Protein polybromo-1 (PB1), also known as BRG1-associated factor 180 (BAF180), plays a protective role in HSC cell senescence through counteracting the p21 transcription factor. The deletion of BAF180 in murine HSCs exhibited deleterious effects on the long-term regeneration and differentiation potency of HSCs [40].

Other intrinsic pathways involved in HSC senescence pathways include p16, JAK/STAT, NF-κB, the mammalian target of rapamycin (mTOR) pathways, the TGF-β signaling pathway, the Wnt pathway, and reactive oxygen species (ROS), among others [24] (Table 1). Targeting cell cycle-dependent kinase inhibitors (CKI) such as p21, mTOR, and p38 mitogen-activated protein kinase (MAPK) was reportedly able to successfully counteract cell cycle defects in order to prevent HSC exhaustion and senescence and to rejuvenate hematopoiesis in elderly mice [41,42,43]. 

#### 2.1.3. Epigenetic Regulation and HSC Ageing 

The loss of epigenetic regulation (such as DNA methylation, histone modifications, and noncoding RNA) is also involved in HSC ageing. Amplification in the methylation peak at the promotor region of genes associated with HSC self-renewal and differentiation capacity contributes to HSC biological dysfunctions [14,62]. DNA methyltransferase enzymes (Dnmt1, Dnmt3a, and Dnmt3b) are responsible for de novo DNA methylation and for maintaining and regulating DNA methylation islands throughout the lifespan [62,63,64,65]. Dntm1 plays a coordinating role to balance between self-renewal and lineage differentiation output in HSCs [63]. The deletion of Dnmt1 resulted in a dramatic reduction in HSC self-renewal and skewed HSC differentiation towards myelopoiesis [63,64,65]. HSCs with Dntm3a^−/−^ and Dntm3b^−/−^ exhibited normal cell expansion and proliferation but showed noticeable deficits in reconstitutional and differentiation capacities in serial transplantation [63,65,66].

Histone posttranslational modifications are crucial in regulating the access of genes that encode proteins to determine HSC fate. Emerging evidence shows that open chromatic mark H3K4me2 is highly expressed in committed and differentiated HSCs, while it is inhibited and downregulated in long-term (LT) HSCs and short-term (ST) HSCs [67]. Transcriptomic and epigenetic analyses have identified a group of chromatin remodeling genes including Kdm3a–b, Kdm5b–d, Jarid1b, and Kdm6a–b) that are age-regulated in HSCs [68,69,70]. For example, it was recently shown that Kdm5b or Jarid1b are involved in regulating the expression of genes such as Hoxa7, Hoxa9, Hoxa10, Hes1, and Gata2 that preserve the self-renewal and proliferation capacities of HSCs [62,71,72]. More evidence has shown that Jarid1b activity declined in aged murine HSCs, and the downregulation of Jarid1b is strongly associated with the upregulation of cell fate–associated genes upon lineage commitment [73]. Additional studies are needed to gain a more comprehensive understanding of the role of chromatin modifications and epigenetic regulations in HSC biology. 

#### 2.1.4. Mitochondria and HSC Ageing 

Another key factor in HSC ageing that remains incompletely characterized is the ageing phenotype at the cellular organelle level [74]. Mitochondria are the major cellular organelles responsible for generating higher levels of energy through the TCA cycle and oxidative phosphorylation [75,76]. Fetal and neonatal HSCs exhibited higher mitochondrial membrane potential (ΔΨmt) and increased mitochondrial dynamics and net mitochondrial mass compared to aged HSCs [77,78,79,80]. ROS and stress responses activate the mammalian sirtuin family SIRT3. SIRT3, also known as the mitochondrial stress regulator gene, triggers mitochondrial protein acetylation, increases the expression of mitochondrial antioxidant enzymes, and enhances ROS scavenging. Recent studies have found that SIRT3 is downregulated in old murine HSCs, which leads to the accumulation of ROS and a decline in mitochondrial plasticity, with a resultant impairment of HSC reconstitutional potency. These findings suggest that SIRT3 is a possible cause of ageing-associated changes in HSCs [81,82].

### 2.2. Extrinsic Factors Driving HSC Ageing

In addition to intrinsic factors, extrinsic factors in the BM microenvironment niche could also affect HSC ageing. These extrinsic factors include cellular components of BM niches, such as hematopoietic cells (megakaryocytes, regulatory T cells, and macrophages) and non-hematopoietic cells (mesenchymal stromal cells, endothelial cells, perivascular cells, and nerve fibers), as well as non-cellular components such as growth factors, inflammatory cytokines and chemokines, and other soluble factors [74,83,84]. These components create an HSC extracellular matrix (ECM) and provide support and crosstalk with HSCs. Any type of pathological or physiological change in the BM microenvironment may impair the steady state of HSCs. For example, injury or stress on BM microenvironmental cells activates the local inflammatory response, resulting in changes in the composition and concentration of local cytokines and interleukins, with important consequences to the biological functions of HSCs. He et al. showed that tumor necrosis factor alpha (TNFα) was sufficient to activate the ERK-ETS1-IL27Rα pathway and push HSCs towards replicative stress and myeloid-biased differentiation [83]. Interferon gamma (INF-γ), macrophage colony-stimulating factor (M-CSF), and gram-negative bacterial component lipopolysaccharide (LPS) are able to interact with HSC innate immune receptors and upregulate several signaling pathways related to HSCs’ biological functional states [85,86,87]. Using single cell RNA sequencing and genome-wide transcriptomic analysis, scientists identified BM niche adhesion factors and stem cell factors such as Ang-1, TPO, Wnt, NOTCH, OPN, and chemokine stromal cell-derived factor-1α (SDF-1α) as factors that directly regulate the steady and differentiation states of HSCs [88]. Tumor growth factor (TGF-β) and Interleukin 6 (IL-6) are overexpressed by aged BM stroma and potentially influence the expression of genes that are associated with HSC ageing. The downregulation of TGF-β and IL-6 had a restorative effect on aged HSCs [89]. Plasma cells are particularly enriched in BM stroma as mice age and express genes associated with inflammatory cytokine response, which is thought to shift HSCs towards ageing myelopoiesis [90]. BM mesenchymal cells from aged mice have a diminished level of insulin-like growth factor 1 (IGF-1). IGF-1 supports HSC survival, and the overproduction with the IGF-1 ligand was able to restore the reconstitution capacity of aged HSCs and mitigate the functional myeloid lineage bias of aged HSCs [91].

BM vascular and endothelial cells provide an important niche for HSCs. Recent studies demonstrated important roles of several BM endothelial transcription factors (EPCR/PAR1 signaling, *Cdc42*, *Ccr9*, *Gnrh2*, and *Lep*) in facilitating the retention and repopulation capacity of HSCs and in reducing the oxidative damage response [92]. CD44 is a member of cell adhesion molecule families expressed by BM niche cells and is crucial in regulating the migration, homing, and survival of HSCs [93]. The deletion of CD44 in neonatal BM ameliorates the engraftment and homing of HSCs [93]. Klf5, a member of the Kruppel-like family which is mostly expressed by BM epithelial cells, plays a pivotal role in enhancing the homing, retention, and lodging of BM HSCs. Klf5 was found to correlate with age-related changes in HSCs [94,95,96].

In summary, the ageing of HSCs is associated with a complex phenotype with characteristic changes from the organelle level to the BM microenvironment. There is a great clinical significance of these changes, as the impairment of HSC functionality such as the loss of cell quiescence is one of the hallmarks of the increased incidence of myeloid and lymphoid leukemias as well as other hematological malignancies with ageing. The impairment of immune function (immunosenescence) contributes to an increased susceptibility to infection and an increased incidence of autoimmune disease. The myeloid skewing has been associated with spontaneously increased levels of proinflammatory cytokines/chemokines and may contribute to frailty and other chronic disease processes in the elderly.

## 3. Reactive Oxygen Species and HSC Ageing

Reactive oxygen species (ROS), such as superoxide anion (O_2_^•−^), hydrogen peroxide (H_2_O_2_), and hydroxyl radical (HO^•^), are the intermediate products of the cellular respiration process which is necessary for metabolic survival in all aerobic organisms. ROS are generated during mitochondrial oxidative phosphorylation (OXPHOS) or during the cellular response to oxidative stress in states such as infection or inflammation. ROS can be thought of as a double-edged sword: a low cellular ROS level plays a critical role in promoting cellular biological functions such as cell proliferation and cell survival [97,98,99,100]. ROS also serve as a critical signaling molecule regulating the mitogen-activated protein kinase (MAPK) pathway, the phosphoinositide 3-kinase (PI3K) pathway, the Nrf2 and Ref1-mediated redox cellular signaling pathway, and the Shc adaptor protein family, among others [101]. An increased ROS level due to the failure of the cellular antioxidant system to scavenge ROS or the overproduction of ROS shifts the cell to a status that is chemically known as oxidative stress. Oxidative stress in the cytosol or in the mitochondria can have deleterious effects on the cell biological systems [102], causing damage to cellular proteins, nucleic acids, lipid membranes, and organelles (in particular, the mitochondria).

### 3.1. Metabolic Status and ROS Production in HSCs

It is well established that oxidative stress has been invariably linked to HSC ageing [103,104,105]. At baseline, HSCs are quiescent and are predominately located in the BM niche with low oxygen tension (physiologic local hypoxia). The quiescent environment of HSCs is further supported by BM osteoblasts that also require low oxygen levels and provide HSCs long-term protection from ROS-related oxidative stress [79]. HSCs transition through different microenvironmental niches during normal physiologic conditions, and, in certain states of hematologic stress such as hemorrhage, viral infection, and radiation injury, HSCs mobilize and migrate from the hypoxic environment to the BM blood sinusoidal system with a highly oxygenic vasculature to increase their catalytic pathways and undergo highly active proliferation and differentiation to restore hematopoietic homeostasis [106,107,108].

When HSCs divide or expand, there is an increased production of ROS due to the change in metabolism. HSCs residing in low oxygen environments mostly generate their fuel through anaerobic glycolysis mediated by pyruvate dehydrogenase kinase 1 (PDK1) and suppress the influx of glycolytic metabolites into mitochondrial membranes. The increased anaerobic glycolysis and suppressed cellular respiration results in the low production of ROS, which promotes cell quiescence, cell renewal, and survival [109,110]. On the other hand, when HSCs are committed to lymphoid and erythroid lineage differentiation, HSCs increase their metabolic flux through pyruvate cycling to significantly increase ATP production, which in turn results in higher levels of ROS [86,88]. This may serve as a feedback loop in which ROS signal a switch of HSC metabolic pathways towards oxidative phosphorylation, with downstream implications for HSC exhaustion and HSC ageing.

Recently, it has been reported that dormant cells such as HSCs generate ROS through the phagocyte NADPH oxidase (NOXs) enzymatic system [111,112,113]. ROS produced by NOXs regulate and contribute to a variety of HSC biological functions. NOX1, 2, and 4 are expressed in human and murine HSCs [114,115]. Interestingly, NOX1 and NOX2 are overexpressed, while NOX4 is downregulated in committed and differentiating HSCs [113,116,117]. More importantly, it has been shown that aged murine HSCs exhibited a global increase in ROS, and this is positively associated with the upregulation of NOX4 [115].

### 3.2. Responses to Oxidative Injury

HSCs have various safeguard mechanisms against the cytotoxic accumulation of ROS. Ataxia telangiectasis mutated (ATM) functions as a sensor of the level of oxidative stress and phosphorylates several key DNA damage response molecules to protect HSCs from oxidative DNA damage. Ito et al. found that the ATM regulation of oxidative stress is crucial for the reconstitutional capacity of HSCs [118]. *Atm*^−/−^ mice older than 24 weeks had elevated ROS levels and showed progressive bone marrow failure due to defects in HSC function. Treatment with anti-oxidant agents restored the reconstitutional capacity of *Atm*^−/−^ HSCs [118].

The mammalian sirtuin family members (SIRTs) play important roles in the regulation of oxidative stress response and have age-specific effects on HSCs. For example, SIRT7^−/−^ mice showed a greater than 40% reduction in the erythroid and myeloid progenies. Furthermore, SIRT7^−/−^ HSCs displayed a dramatic decline in the long-term reconstitution potency [80,119]. Aged HSCs exhibited SIRT7 downregulation compared to neonatal and young HSCs. The deletion of SIRT6 promotes HSC stress replication through the aberrant activation of the Wnt signaling pathway. The reintroduction of SIRT3 into aged murine HSCs improves their mitochondrial plasticity and regenerative capacity [80,82,119,120].

PI3K-AKT-mTOR-reactive oxygen species-p53, MDM2, ATM, MAP K, p38, p16, and p21 have all been identified as oxidative/antioxidant regulatory genes [121]. Responding to oxidative injury involves a complex process characterized by the dephosphorylation of p53/MDM2 and the activation of the p53 apoptotic signaling pathway [121,122]. Forkhead box class O family member proteins (FoxOs) including FOXO1, FOXO3a, and FOXO4 are essential transcription factors that enhance a return to quiescence and the survival of murine and human HSCs in response to oxidative stress and damage [123]. Free radicles and oxidative stress cause the phosphorylation of FOXO1, FOXO3a, and FOXO4, which then activates a group of genes involved in programmed cell death and cell cycle regulation [123]. FoxO triple negative mice exhibited a severe decline in HSC integrity and increased myeloid differentiation bias. In this case, administration of N-acetyl cysteine (NAC) could only partially rescue HSC defects, which suggested that ROS might not be the main cause of HSC deficiency seen in FoxO triple negative mice [123]. Hypoxia-inducible factor-1α (HIF-1α) shifts cellular metabolism from mitochondrial respiration to glycolysis and reduces ROS production. Cellular anti-oxidant systems such as the superoxide dismutase, glutathione system, and thioredoxin system can scavenge ROS and reduce the level of ROS. Mitochondrial superoxide dismutase (SOD2) or cytosolic SOD1 converts superoxide radical anion (O_2_^•−^) to hydrogen peroxide (H_2_O_2_), and Glutathione-dependent enzymes catalyze biological redox reactions and assist in the protection against ROS and oxidative damage.

### 3.3. Oxidative Injury and HSC Ageing

The imbalance between ROS production and the capacity of anti-oxidant systems to counteract ROS results in oxidative stress. Oxidative stress induces DNA base damage and causes the release of free bases and the generation of a basic site, leading to increased cell cycling and apoptosis as well as compromised self-renewal and the differentiation of HSCs [43,118,124]. HSCs in the hypoxic BM niche re-localized HIF-1α to the nucleus to promote a plethora of genes that are involved in maintaining the HSC quiescent cell cycle, cell survival, self-renewal, and appropriate mitochondrial mass. Increases in the ROS level in HSCs induces the degradation of the HIF-1α protein and activates a group of genes that are associated with cell commitment and differentiation [125,126]. Nuclear factor erythroid 2–related factor 2 (Nrf2) has emerged as a master transcription factor that regulates a plethora of antioxidant genes in HSCs. The conditional deletion of Nrf2 in mouse models increases HSC sensitivity to ROS and impairs the regenerative capacity of HSCs, which could not be rescued by the administration of NAC [127]. Oxidative stress may directly trigger a number of stress response signaling pathways such as p38 MAPK kinase and limit the lifespan of HSCs. The p38 MAPK signaling pathway is involved in the activation of cellular apoptosis, autophagy, and cellular differentiation [43]. In *Atm*^−/−^ mice, increased ROS induces the HSC-specific phosphorylation of p38 MAPK and defects in the maintenance of HSC quiescence. Prolonged treatment with an antioxidant or an inhibitor of p38 MAPK extended the lifespan of HSCs from wild-type mice in serial transplantation experiments [43]. Several studies have found that the thioredoxin interacting protein TXNIP/p53 axis plays a role in rescuing and reconstituting exhausted HSCs under oxidative stress [128]. Exposing HSCs to oxidative damage or high metabolic rate demands disturbs the intermolecular disulfide interaction between TXN1-TXIP, resulting in TXN1 nuclear translocation to regulate cell survival and growth through the ERK1/2 MAPK signaling pathway [128,129,130]. The TXNIP-p38 axis was found to be a promising target in rejuvenating aged murine HSCs [131,132]. ROS can also limit the ability of bone marrow stromal cells to support hematopoietic reconstitution [133].

In summary, ROS concentrations can be thought of as a sensor of the steady state and stemness capacity of HSCs in various signaling pathways, and the accumulation of oxidative stress damage may trigger several cell events including apoptosis, autophagy, and differentiation, which are associated with HSC ageing.

## 4. The Utility of Antioxidants in the Biology of HSC Ageing

Antioxidants are molecules or proteins with an ability to modulate cellular oxidation reactions by donating an electron to reduce or prevent the oxidation of oxidizable molecules.

Endogenous antioxidants are categorized into non-protein antioxidant molecules and antioxidant proteins. Non-protein antioxidant molecules are glutathione, alpha-lipoic acid (LA), coenzyme Q (CoQ), ferritin, uric acid, and bilirubin. Key protein antioxidants include superoxide dismutase (SOD), catalase (CAT), thioredoxin, and glutathione peroxidase. Endogenous antioxidants are the first line of the cellular defense mechanism against oxidative damages [134,135,136]. The thioredoxin and glutathione systems are the two major mammalian endogenous antioxidant systems. The thioredoxin and glutathione systems work in parallel to reduce oxidative metabolites and to supply a steady state of reduction power in the cytosolic environment to protect cells from the exposure to harmful levels of ROS [137,138]. There are a variety of exogenous antioxidant agents such as phenolic compounds, ascorbic acid, tocopherol, N-acetyl cysteine, and curcumin. Pharmaceutical companies have developed and are further developing drugs to work as antioxidants or synergize with antioxidant systems to prevent/delay the ageing process or to rejuvenate aged cells [138,139,140].

Although oxidative stress and oxidative damage have been established to be highly associated with the ageing process in a wide variety of organisms, the causative relationship between oxidative stress and ageing remains unclear. In cell culture systems and in some animal models, the administration of antioxidants has been shown to rejuvenate aged HSCs and extend the lifespan of HSCs [43,118,141]. The protective effects of antioxidants against free radicles have attracted many scientists to investigate whether antioxidants can be used to reverse ageing-associated functional decline and to improve the cellular lifespan and longevity. The results have thus far been mixed. In radiation injury models, antioxidant agents have been shown to decrease oxidative stress and improve the recovery of hematopoiesis following radiation injury, but these substances have not improved the regenerative capacity of HSCs nor shown restorative effects on aged HSCs. Radiation-induced HSC damage is mechanistically distinct from age-associated changes in HSCs [100,102,142]. Clinical trials with the supplementation of antioxidants did not prove beneficial in reducing ageing-associated disease or improving life expectancy. Further complicating the picture, it has been shown that the prolonged exposure to the exogenous antioxidants could have deleterious effects on the endogenous antioxidant systems, and some studies have shown that the long-term introduction of antioxidants in the diet actually reduced the overall lifespan [143,144].

Studies in knock-out and transgenic mice intended to manipulate antioxidant enzymes, including Cu/Zn superoxide dismutase (SOD), MnSOD, catalase, and glutathione peroxidase, have largely failed to show changes in the lifespan of the animals despite the exposure to extreme levels of oxidative stress [141,145]. Dyskeratosis congenita (DC) is a genetic disorder disease characterized by the loss of telomerase activity and the accumulation of DNA damage and is associated with reduced HSC regeneration ability and the decline in several HSC biological functions, the hallmark of HSC ageing. Using the DC mouse model (Dkc1^Δ15^), researchers found that the ageing-associated changes in HSCs were associated with high concentrations of ROS. However, the administration of NAC to Dkc1^Δ15^ mice for one year was not able to reverse ageing-associated changes in HSCs [146].

In summary, HSCs have complex, highly conserved, and multi-faceted endogenous antioxidant systems that protect and enhance HSC recovery in response to ROS and oxidative stress injuries. Exogenous antioxidants and synergistic agents augment the efficiency of the cellular endogenous systems, although this effect is likely transient and has not been shown to definitively produce a longevity benefit. Genetically engineered mouse models for the majority of antioxidant enzymes have not demonstrated changes in the lifespan of the animals.

## 5. Role of Thioredoxin in HSC Ageing Biology

### 5.1. Overview of the Thioredoxin System

The thioredoxin system is one of the major cellular antioxidant proteins in mammals and is highly evolutionarily conserved amongst aerobic organisms. The thioredoxin system consists of thioredoxin (Trx1), thioredoxin reductase (TrxR), and nicotinamide adenine dinucleotide phosphate (NADPH as an electron donor to recycle oxidized Trx to its reducing form). Ultimately, the thioredoxin antioxidant system catalyzes electron flux from NADPH through TrxR reductase to Trx, which then keeps cellular organic molecules (proteins, lipids, and DNAs) in the reduced form (Figure 3). The thioredoxin system is not only responsible for reducing proteins but also for restoring proteins to their native structure, thus providing resistance against stress and maintaining cellular integrity [147].

Thioredoxin (Trx) has two main isoforms: Trx-1 and Trx-2. Trx-1 is the predominant isoform and is primarily cytosolic but easily crosses cell and nuclear membranes [137]. Trx-1 translocates into the nucleus upon stress conditions or can be secreted extracellularly via a unique leaderless mechanism. Trx-2 is restricted in mitochondria [137]. Our current review focuses on Trx-1. Trx-1 is a 12-kD target-selective protein disulfide reductase with two Cys residues in its conserved active site. Trx is a multifunctional protein and has a distinct structure that facilitates this translocation to the nucleus in certain conditions as well as exports to the extracellular membrane to serve as proinflammatory cytokines [148,149]. Trx-1 provides a reducing equivalent that sustains a variety of cell biological functions, including immune cell survival and cell proliferation, and maintains cellular redox homeostasis [150]. Compared to other known reducing systems in the cell, Trx-1 is the only protein that maintains the reducing power for the ribonucleotide reductase enzyme, which is the building block for DNA replication and repair [150,151]. In addition to functioning as antioxidant, the reduced form Trx-(SH)2 contains a dominant motif to catalyze proteins in a manner similar to protein phosphorylation and dephosphorylation and can interact with a variety of proteins, including transcription-binding factors at the genomic level [152,153,154].

### 5.2. Trx-1: One of the Few Antioxidants Shown to Extend Lifespan in Transgenic Mouse Models

The role of Trx-1 in maintaining cellular functions such as cell growth, cell survival, and cell longevity has been extensively studied but remains incompletely defined. The Kopf lab found that the thioredoxin system is critical for maintaining reducing power for the synthesis of ribonucleotides critical to the process of T cell DNA biosynthesis [155]. The inactivating components of the thioredoxin system resulted in the impaired development and activation of T cells and myeloid populations and increased cell replication stress [155,156]. Furthermore, studies have investigated the association between Trx-1 and lymphoid homing, migration, killing abilities, and cytotoxic granules release by NK cells [157]. These studies demonstrated a significant role of Trx-1 in maintaining the health of both hematopoietic and immune systems throughout the ageing process [155,156,157].

With greater relevance to this review, the role of Trx-1 in biologic ageing and longevity has been investigated by the Yodoi lab, who have successfully generated thioredoxin (encoded by Trx1 gene) transgenic mice (Tg, act-TRX1^+/0^) by overexpressing human Trx1 cDNA driven by the β-actin promotor [158]. Interestingly, these Trx-1 Tg mice have shown a 35% extension in the lifespan compared to the matched controls [159], and tg mice are more resistant to various inflammatory and immune injuries [160,161,162]. Furthermore, Trx-1 Tg mice have shown higher telomerase activity, widely considered as a substitute for a “biological clock”. These data suggest that Trx-1 has biologic functions beyond the well-described redox homeostasis functions and may play an important role in processes such as stem cell differentiation/renewal, cellular senescence, and the lifespan of the organism.

The Ikeno lab published the first long-term survival study to further examine the effects of overexpressing Trx-1 on ageing and ageing-associated pathology. The Ikeno team cloned the same human Trx-1 transgene described above to generate TRX1 transgene mice (Tg, act-TRX1^+/0^), but with a longer follow-up (29–30 months) under optimal conditions. They found that the overexpression of TRX1 has benefits at both the cellular and organ levels, and the lifespan extension occurs mainly in earlier ages with limited effects in later ages. Additionally, young Trx-1 transgenic mice showed noticeable improvements in immune function, inflammatory response, and the resistance to oxidative damage compared to the control group. It was also noted that aged Trx-1 transgenic mice have shown a higher incidence of total fatal tumors and fatal lymphoma compared to wild-type mice [163,164].

These findings demonstrated an important role of TXN1 in ageing, and Trx-1 is one of the very few antioxidants that are found to extend the lifespan of transgenic mice.

### 5.3. Thioredoxin-1 Enhances HSC Functions in Animal Models of Hematopoietic Stem Cell Transplant and Radiation Injury

The role of Trx-1 in the function and ageing of HSCs remains unclear. Using a mass-spectrometry based semi-quantitative proteomics screen, our lab previously showed that Trx-1 was significantly upregulated in the BM of hematopoietic stem cells transplanted (HSCT) recipient mice treated with AMD3100, also known chemokine receptor type 4 (CXCR4) antagonist (plerixafor, trademark Mozobil), relative to the controls [165]. AMD3100 treatment has improved hematopoietic recovery following myeloablative HSCT in our mouse model and in patients receiving myeloablative allogeneic transplants [165,166]. We have demonstrated the marked proliferative and protective effects on HSCs in animal models of HSCT and radiation injury. We showed that the ex vivo culture of murine HSCs with recombinant Trx-1 enhances their long-term repopulation capacity, reduces cell senescence and radiation-induced double-strand DNA breaks, and downregulates apoptotic-signaling pathways. We found that the administration of recombinant Trx-1 up to 24 h following lethal total body irradiation (TBI) rescues BALB/c mice from radiation-induced lethality [167].

It remains to be determined how AMD3100 increases Trx-1 expression. AMD3100 may increase Trx-1 indirectly through other mediators in the niche microenvironment. For example, studies have shown that AMD3100 attenuates the expression of Txnip, which directly binds to the reduced form of Trx-1 and inhibits Trx-1 expression and activity [168,169]. Txnip^−/−^ mice exhibited the overexpression of CXCR4/SDF-1, which was downregulated by the administration of AMD3100 [168,169].

### 5.4. Emerging Evidence of Thioredoxin-1 in Protecting HSCs from Ageing

The protective effects of Trx-1 on HSCs in HSCT and in radiation injury led us to determine the role of Trx-1 in HSC ageing. We first sorted by flow cytometry Lin^−^Sca-1^+^c-Kit^+^CD150^+^CD48^−^ long-term (LT)-HSCs, Lin^−^Sca-1^+^c-Kit^+^CD150^−^CD48^−^ short-term (ST) HSCs, Lin^−^Sca-1^+^c-Kit^+^CD150^−^CD48^+^ multipotential progenitors (MPPs), and lineage+ cells from C57BL/6J mice aged at 3 weeks, 27 weeks, or 65 weeks; then, we measured Trx-1 and p53 mRNA expression. Trx1 mRNA expression dramatically increases in the LT-HSCs from 65-week-old mice compared to that from mice at a younger age. *TP53* mRNA expression noticeably reduced in the aged mice compared to the younger mice (Figure 4A,B). To test whether Trx-1 plays a critical role in regulating HSC ageing, our lab crossbred TXN1 conditional knockout mice (TXN^fl/fl^) [170] with ROSA-ER-Cre mice and generated ROSA-ER-Cre-TXN^fl/fl^. ROSA-Cre-TXN^fl/fl^ mice and control mice (either TXN^fl/fl^ or ROSA-ER-Cre mice) were treated with tamoxifen (75 mg/kg i.p. daily for 5 days), which leads to the complete knockout of TXN1 in bone marrow samples at day 10 of injection. Our unpublished data suggested that the deletion of TXN1 significantly attenuated the reconstitutional, self-generating, and repopulating capacities of HSCs in lethally irradiated recipient mice. Furthermore, HSCs lacking TXN1 have shown phenotypes characteristic of aged HSCs, including the higher expansion and proliferation of LT-HSCs (unpublished data). We measured the expression of genes that are associated with the ageing phenotypes of HSCs, such as p16, p21, p38/MAPK, and Wnt5a [36,171]. The deletion of TXN1 in HSCs upregulated the mRNA expression of p16, p21, p38/MAPK, and Wnt5a (Figure 4C). We measured the peripheral leukocytes in the ROSA-Cre-TXN^fl/fl^ mice and control mice at different ages (2 months and 10 months old). The percentage of leukocytes in the peripheral blood cells reduced by more than twofold after the deletion of TXN1 in young (2 months old) mice compared to the age-matched control animals. No significant change in the percentage of peripheral blood leukocytes was observed in old (10 months old) mice (Figure 4D). Our data suggested that the role/contribution of Trx-1 in HSC ageing could be different at different stages of the lifespan.

### 5.5. Thioredoxin-1 Mediated Signaling Pathways in HSCs

Through ROS dependent and independent mechanisms, thioredoxin plays an important role in apoptotic regulation. In studies of murine thymoma cells (WEHI7.2), high endogenous and exogenous Trx-1 was shown to be protective against both spontaneous and drug-induced apoptosis [172]. Apoptosis regulator ASK1, a member of the MAPKK family, activates apoptosis through a JNK-dependent pathway. A redox reaction involving Trx-1 has been shown to play a crucial role in this apoptotic pathway, with the overexpression of Trx1 resulting in the increased reduction of ASK1, the impaired activation of JNK, and, ultimately, decreased apoptosis with a minimal impact on the levels of ASK1 activation [173]. The reduced form of Trx-1 directly interacts with the N terminal of ASK1 and inhibits its kinase activity. Oxidative damage leads to oxidized Trx1, which dissociates ASK1, and ASK1 then activates the SEK1-JNK and MKK3/MKK6-p38 signaling cascades to induce program cell death. Additionally, Trx-1 promotes ASK1 ubiquitination and degradation, and the association of Trx1/ASK1 could not be blocked by ROS or other apoptotic signaling stimuli [173]. Trx-1 regulates NF-kappa B DNA binding affinity by the reduction of a disulfide bond involving cysteine 62. Several lines of evidence have found that Trx1 regulates p53 activity.

### 5.6. Thioredoxin 2 and Mitochondrial Contributions to HSC Ageing

Recent studies have highlighted the significance of the mitochondrial regulation of HSC quiescence [174]. Increase mitochondrial biogenesis is a key feature of the transition of HSCs from quiescence to proliferation due to the increased energy requirement of active HSCs. There is emerging evidence for an important mitochondrial checkpoint in the cell cycle regulation of quiescence. Several mechanistic studies have begun to describe an mTOR-LKB1-AMPK-dependent pathway regulation mitochondrial biogenesis, mitochondrial mass, and mitochondrial activity in the G_0_-to-G_1_ cell cycle transition [80,175,176]. In the reverse order, it has been suggested that inadequate mitogenesis or a mitochondrial stress response (with contributing mitophagy) leads to a loss of HSC quiescence. If mitochondrial stress is not adequately repaired, cell death occurs through apoptosis in physiologic states. Although this review focuses primarily on Trx-1, Trx-2 is a mitochondrial-restricted isoform of Trx with additional importance to HSC redox biology and ageing. The specific inactivation of murine Trx2 is embryonically lethal in mice. Heterozygous deletion results in a phenotype characterized by anemia, hepatic apoptosis, and significantly reduced ex vivo hematopoiesis [177]. This finding suggests a critical role for the thioredoxin system in HSC function and fitness through a mitochondrial restricted pathway related to mitochondrial-mediated apoptosis, as described above. This mechanism has been further characterized: in glutathione-depleted cells, Trx2 becomes oxidized, inactivated Trx2 increases the susceptibility to TNF-alpha and ROS, and the overexpression of Trx2 protects against this toxicity [178,179,180]. Another unique aspect of mitochondrial Trx2 is that Trx2 detoxifies ROS through mitochondrial specific peroxiredoxins (Prx3) and has its own reductase (TrxR2) [179,180,181].

In sum, the thioredoxin system consists of functionally distinct but synergistic cytosolic Trx1 and mitochondrial Trx2 pathways, both regulating ASK1 activity and apoptosis.

## 6. Conclusions

As with all cells, HSCs regulate cellular functions through the apoptosis and senescence pathways, with important implications for HSC ageing. As discussed above, a key driver of HSC ageing is the accumulation of DNA damage through ROS exposure and DDR errors. There is an evolutionary pressure to tightly balance the risk of malignant transformation to hematologic malignancies with the importance of maintaining adequate hematopoiesis throughout the lifespan. Complex cascades of molecular events and pathways including DNA damage response, ROS, mitochondrial regulatory mechanisms, and apoptotic pathways work in concert to maintain this balance. Our data suggest that thioredoxin appears to play an important role in maintaining the biological functions of HSCs. These findings provide a rationale and justification for further investigating the effects of TXN1 as a molecule for mitigating against ageing-related changes in HSCs.

## Figures and Tables

**Figure 1 antioxidants-11-01291-f001:**
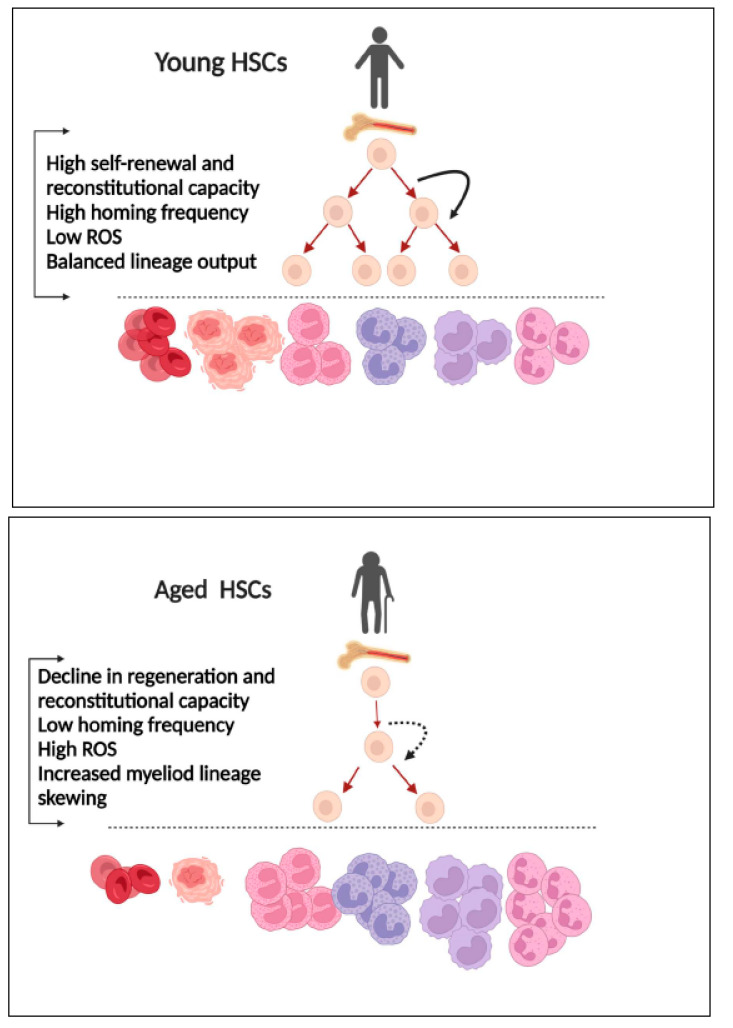
Ageing-associated changes in HSCs. Young HSCs exhibit high self-renewal and reconstitutional capacities, a high homing frequency, low ROS, and a balanced lineage output. Aged HSCs are characterized by a decline in self-renewal and reconstitutional capacities, high ROS, a low homing frequency, and increased myeloid lineage skewing. Created with BioRender.com, accessed on 19 June 2022.

**Figure 2 antioxidants-11-01291-f002:**
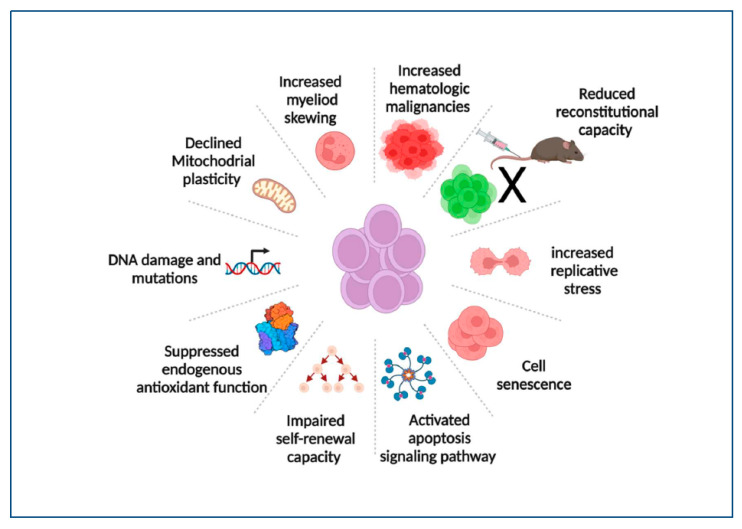
Hallmarks of HSC ageing. Aged HSCs exhibit impaired self-renewal and reconstitutional capacities and increased replicative stress. They are also characterized by cell senescence, increased cell apoptosis, and declined mitochondrial biological functions. Aged HSCs are associated with a deficiency in the DNA repair pathway, higher mutation rates, increased myeloid lineage skewing, and an increased incidence of hematological malignancies. Created with BioRender.com, accessed on 19 June 2022.

**Figure 3 antioxidants-11-01291-f003:**
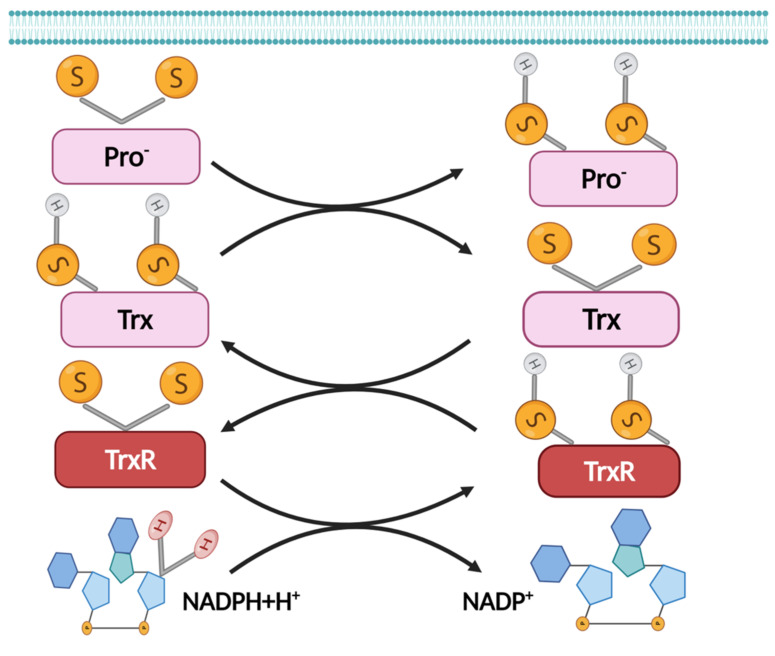
Thioredoxin system. Trx in the oxidized form, Trx-S2, or the reduced form, Trx-(SH)2. In the reduced state, Trx directly reduces disulfides in oxidized substrate proteins (Pro^−^-S2). The oxidation reaction is reversible and is maintained by thioredoxin reductase TrxR, which is sustained by the electron donor NADPH. Created with BioRender.com, accessed on 19 June 2022.

**Figure 4 antioxidants-11-01291-f004:**
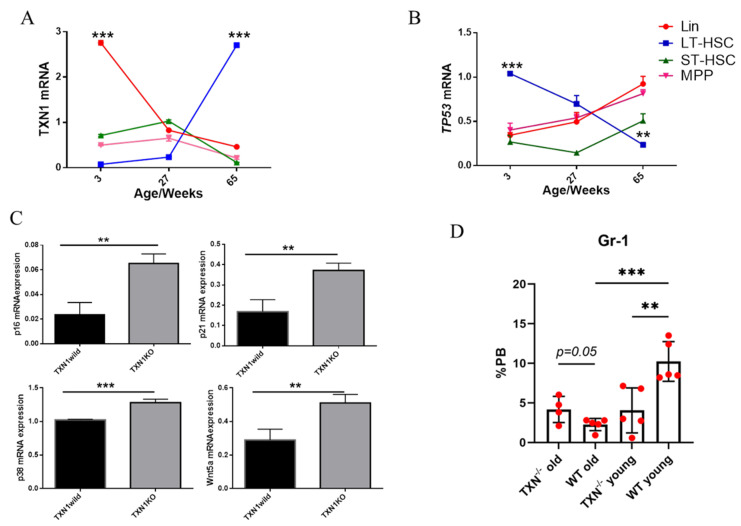
The loss of TXN1 enhances hematopoietic stem cells ageing: (**A**,**B**) Various hematopoietic cell populations from wild-type mice C57Bl/6 at 3, 27, and 65 weeks of age were sorted and measured for TXN1 (**A**) and *TP53* (**B**). (**C**) BM Lin cells from the ROSA-ER-Cre-TXN^fl/fl^ mice and controls were used to measure mRNA expression for Wnt5a, p21, p16, and p38. (**D**) The ROSA-ER-Cre-TXN^fl/fl^ mice and controls at 2 months or 10 months were treated with tamoxifen, and at 10 days, peripheral leukocytes were measured. The data are represented as the mean ± SD. ** *p* < 0.01, *** *p* < 0.001.

**Table 1 antioxidants-11-01291-t001:** Selected molecules and pathways in the regulation of HSC senescence.

Cell Senescence Molecules and Pathways	Functional Activities	References
p53-p21 axis	Telomerase activity Oxidative stress Terminal cell cycling arrest	[44]
EZh1 and EZh2, also known as polycomb protein members	Differentiation Self-renewal Genomic integrity	[45]
SA-β-galactosidase (SA-β-Gal) and lipofuscin	Clonogenic capacity Oxidative stress DNA damage	[46]
Bmi1, a member of the Polycomb group proteins	Reconstitution, repopulation, and self-renewal capacities Mitochondrial production of ROS	[47,48]
Ink4a/Arf transcription factors	Apoptosis Cell cycle arrest Senescence-associated heterochromatic foci (SAHF)	[49,50]
Arf/P53 pathway	HSC expansion and self-renewal Apoptosis Exhaustion and stress proliferation response	[51,52,53]
p38/MAP kinase signaling pathway	DNA damage Oxidative stress Telomerase activity Exhaustion and stressful replication	[54,55,56]
Ataxia-telangiectasia mutated (ATM) and Telomerase reverse transcriptase (TERT)	Oxidative stress Biological functions Self-renewal capacity Apoptosis	[57]
Chemokines and cytokines including IL-8 (CXCL8), GROα (CXCL1), GROβ (CXCL2), GROγ (CXCL3), MCP-1 (CCL2), MCP-2 (CCL8), MCP-4 (CCL13), MIP-1α (CCL3), MIP-3α (CCL20), and HCC-4 (CCL16).	Seeding efficiency and homing Differentiation Mobilization and migration Proliferation and expansion	[58,59,60,61]

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
