# Peer review of "Emerging Evidence of the Significance of Thioredoxin-1 in Hematopoietic Stem Cell Aging"

_antioxidants, 2022, doi:10.3390/antiox11071291_

Round 1

Reviewer 1 Report

The authors present a Review entitled “Emerging evidence of the Significance of Thioredoxin-1 in Hematopoietic Stem Cell Aging”. The aim of this manuscript is to summarize and discuss the effect of aging on hematopoietic stem cells (HSCs) and the role of thioredoxin-1 in regulating HSC aging.

General comment

This is a complete review of the topic, which describes the main differences found in HSC of young animal models compared to older ones, and the main factors related to HSC aging. In particular, aged HSC showed reduced reconstitution potency, differentiation kinetic, homing and engraftment efficiency, changes from the organelle level to the specialized bone marrow microenvironment.

However, as reported by the authors, cellular and molecular pathways driving HSC ageing are complex and not clarified, and considering that the alteration of HSC could be the cause of severe haematological disorders such as leukemia, myeloma and lymphoma, this manuscript represents a significant study laying the foundations for new investigation area aimed to understanding the mechanisms underlying aging and for development of new cancer therapies.

The authors deal very well with the role of oxidative stress in HSC aging and protective role of antioxidant molecules, in particular the protective effects thioredoxin system on HSC in radiation damage. The authors present interesting preliminary data on the role played by TXN1 in maintaining the capacity for reconstitution, self-generation and repopulation of HSCs.

For the completeness of the information, the scientific impact of the data presented and the clarity in the drafting of the manuscript this reviewer assumes that the manuscript deserves to be published.

However, this reviewer suggests some revisions prior to publication as follows:

specific comment

line 49: the paragraph is too long, to make it more fluent, it would be advisable to divide it into subparagraphs, as an example: “intrinsic factors related to HSC ageing and extrinsic factors related to HSC ageing”.

Add a schematic overview of the thioredoxin antioxidant system.

Conclusion section should be integrated with evidence of thioredoxin-1 in protecting HSCs from ageing.

Author Response

Title: Emerging evidence of the Significance of Thioredoxin-1 in Hematopoietic Stem Cell Aging

We greatly appreciate the time and effort the reviewers spent reviewing our manuscript and we thank them for their constructive comments and excellent critiques. The manuscript was revised accordingly. The changes were marked in red color in the revised manuscript. The point-to-point responses to the Reviewers’ comments are included below.

Point-to-point responses to reviewers’ comments:

Reviewer#2:

The authors present a Review entitled “Emerging evidence of the Significance of Thioredoxin-1 in Hematopoietic Stem Cell Aging”. The aim of this manuscript is to summarize and discuss the effect of aging on hematopoietic stem cells (HSCs) and the role of thioredoxin-1 in regulating HSC aging.

General comment

This is a complete review of the topic, which describes the main differences found in HSC of young animal models compared to older ones, and the main factors related to HSC aging. In particular, aged HSC showed reduced reconstitution potency, differentiation kinetic, homing and engraftment efficiency, changes from the organelle level to the specialized bone marrow microenvironment.

However, as reported by the authors, cellular and molecular pathways driving HSC ageing are complex and not clarified, and considering that the alteration of HSC could be the cause of severe haematological disorders such as leukemia, myeloma and lymphoma, this manuscript represents a significant study laying the foundations for new investigation area aimed to understanding the mechanisms underlying aging and for development of new cancer therapies.

The authors deal very well with the role of oxidative stress in HSC aging and protective role of antioxidant molecules, in particular the protective effects thioredoxin system on HSC in radiation damage. The authors present interesting preliminary data on the role played by TXN1 in maintaining the capacity for reconstitution, self-generation and repopulation of HSCs.

For the completeness of the information, the scientific impact of the data presented and the clarity in the drafting of the manuscript this reviewer assumes that the manuscript deserves to be published.

However, this reviewer suggests some revisions prior to publication as follows:

specific comment

line 49: the paragraph is too long, to make it more fluent, it would be advisable to divide it into subparagraphs, as an example: “intrinsic factors related to HSC ageing and extrinsic factors related to HSC ageing”.

Response: Yes. The paragraph has been subdivided into intrinsic factors in line 101. Extrinsic factors in line 194.

Add a schematic overview of the thioredoxin antioxidant system.

Response: We have generated schematic figure 3 explained the mechanism of TXN1 system using Biorender software.

Conclusion section should be integrated with evidence of thioredoxin-1 in protecting HSCs from ageing.

Response: We have addressed this in line 550-556.

Reviewer 2 Report

In this review by Jabbar et al, the authors provide an in-depth review of hematopoietic stem cell (HSC) aging, with a specific focus on roles of thioredoxin-1, the only antioxidant protein shown to influence cellular aging. The review begins with an overview of both biochemical and molecular pathways that regulate cellular aging, with a focus on those shown to directly affect HSCs. The different described mechanisms include intrinsic factors as components of multiple signaling pathways, and extrinsic factors that influence or are controlled by the bone marrow microenvironment. The authors then introduce the roles of reactive oxygen species in aging HSCs, which logically leads to a discussion of antioxidants that can influence ROS and oxidative stress. The review then targets thioredoxin, in particular Trx-1, including how increased expression can extend lifespans of transgenic mice, which includes some of their own data on mice with engineered Txn1 overexpression and different HSC subpopulations plus RNA expression analyses. The data is understandable and adequately presented. Overall this is a well-written review that touches on multiple molecular pathways and mechanisms critical to the loss of HSC self-renewal in aging mice and humans, and in particular sheds light on the importance of further studies on Txn-1, and perhaps Txn-2, in HSC longevity and self-renewal capacities. There are some issues that should be considered as follows.

1. Under "2. Background...", lines 100-185, the authors might add subheadings for the different types of molecular mechanisms that cause HSC aging, with associated paragraphs to ease reading of the different mechanisms (as is done for the different sections for thioredoxin). For example, subheading could be added for DNA Damage responses, intrinsic signaling pathways, BM microenvironment changes, etc.

2. Line 136, define SIRT2 as a member of the sirtuins family of histone deacetylases, and how it is related to the comments on mitochondrial stress, for clarity (and added in lines 231-233 later on).

3. Line 189, the authors insert a definition for cellular respiration (assuming this is the purpose of "Box 1") and the definition provided at the end of the text (lines 497-505), but perhaps this can be compressed into a descriptive sentence here? Also, the description has some grammatical errors.

4. Lines 243 - 281, this paragraph should be divided, as it is too long and difficult to navigate - suggest breaking into two sections/paragraphs, the first on effects of ROS on proteins and targeted signaling pathways, and the second on responses to oxidative injury.

5. Line 330, the authors use the acronym NAC, but if used for N-acetyl cysteine, this should be included with this term for the acronym (line 300).

6. Line 405, the authors need to explain how AMD3100 treatment correlates with Trx-1 functions to enhance HSC repopulation/longevity. How does a CXCR4 antagonist effect Trx-1?

7. Line 428-430, the authors might mention that they have evidence that TXN1 deletion attenuates HSC properties, rather than the direct statement of effect until the manuscript is actually accepted for publication.

8. Line 439-440, the sentence needs editing as the end phrase is confusing ("in different stage of lifelong").

9. Line 474, the phrase "leads to return to HSC quiescence" needs editing.

Author Response

Title: Emerging evidence of the Significance of Thioredoxin-1 in Hematopoietic Stem Cell Aging

We greatly appreciate the time and effort the reviewers spent reviewing our manuscript and we thank them for their constructive comments and excellent critiques. The manuscript was revised accordingly. The changes were marked in red color in the revised manuscript. The point-to-point responses to the Reviewers’ comments are included below.

Point-to-point responses to reviewers’ comments:

Reviewer#1:

In this review by Jabbar et al, the authors provide an in-depth review of hematopoietic stem cell (HSC) aging, with a specific focus on roles of thioredoxin-1, the only antioxidant protein shown to influence cellular aging. The review begins with an overview of both biochemical and molecular pathways that regulate cellular aging, with a focus on those shown to directly affect HSCs. The different described mechanisms include intrinsic factors as components of multiple signaling pathways, and extrinsic factors that influence or are controlled by the bone marrow microenvironment. The authors then introduce the roles of reactive oxygen species in aging HSCs, which logically leads to a discussion of antioxidants that can influence ROS and oxidative stress. The review then targets thioredoxin, in particular Trx-1, including how increased expression can extend lifespans of transgenic mice, which includes some of their own data on mice with engineered Txn1 overexpression and different HSC subpopulations plus RNA expression analyses. The data is understandable and adequately presented. Overall this is a well-written review that touches on multiple molecular pathways and mechanisms critical to the loss of HSC self-renewal in aging mice and humans, and in particular sheds light on the importance of further studies on Txn-1, and perhaps Txn-2, in HSC longevity and self-renewal capacities. There are some issues that should be considered as follows.

1. Under "2. Background...", lines 100-185, the authors might add subheadings for the different types of molecular mechanisms that cause HSC aging, with associated paragraphs to ease reading of the different mechanisms (as is done for the different sections for thioredoxin). For example, subheading could be added for DNA Damage responses, intrinsic signaling pathways, BM microenvironment changes, etc.

Response: Yes. We have added subheading in the revised manuscript. Line 97-106, 130-194.

2. Line 136, define SIRT2 as a member of the sirtuins family of histone deacetylases, and how it is

related to the comments on mitochondrial stress, for clarity (and added in lines 231-233 later on).

Response: Yes. SIRT3 functions have clarified in line 185-192. SIRTs family have explained in details under the category of (responses to oxidative injury), line 294-301.

3. Line 189, the authors insert a definition for cellular respiration (assuming this is the purpose of "Box 1") and the definition provided at the end of the text (lines 497-505), but perhaps this can be compressed into a descriptive sentence here? Also, the description has some grammatical errors.

Response: Yes. The descriptions of ROS and cellular respiration were merged in one sentence in line 244-246.

4. Lines 243 - 281, this paragraph should be divided, as it is too long and difficult to navigate - suggest breaking into two sections/paragraphs, the first on effects of ROS on proteins and targeted signaling pathways, and the second on responses to oxidative injury.

Response: Yes. The paragraph have sub headed. Line 257, 297-285, 286, 294-319.

5. Line 330, the authors use the acronym NAC, but if used for N-acetyl cysteine, this should be included with this term for the acronym (line 300).

Response: The acronym of N-acetyl cysteine (NAC) was mentioned in line 312. After that we used symbol NAC referring to N-acetyl cysteine. Line 323, 390.

6. Line 405, the authors need to explain how AMD3100 treatment correlates with Trx-1 functions to enhance HSC repopulation/longevity. How does a CXCR4 antagonist effect Trx-1?

Response: we have addressed the possible mechanisms of AMD3100 in line 474-479.

7. Line 428-430, the authors might mention that they have evidence that TXN1 deletion attenuates HSC properties, rather than the direct statement of effect until the manuscript is actually accepted for publication.

Response: Yes. we have clarified this issue in line 493-497.

8. Line 439-440, the sentence needs editing as the end phrase is confusing ("in different stage of lifelong").

Response: Yes. Sentence was explained in line 503-504.

9. Line 474, the phrase "leads to return to HSC quiescence" needs editing.

Response: The sentence was corrected in line 530.